# Targeted and Checkpoint Inhibitor Therapy of Metastatic Malignant Melanoma in Germany, 2000–2016

**DOI:** 10.3390/cancers12092354

**Published:** 2020-08-20

**Authors:** Peter Hellmund, Jochen Schmitt, Martin Roessler, Friedegund Meier, Olaf Schoffer

**Affiliations:** 1Center for Evidence-based Healthcare, Faculty of Medicine and University Hospital Carl Gustav Carus, TU Dresden, 01307 Dresden, Germany; Peter.Hellmund@tu-dresden.de (P.H.); Jochen.Schmitt@uniklinikum-dresden.de (J.S.); Martin.Roessler@uniklinikum-dresden.de (M.R.); 2Faculty of Medicine and University Hospital Carl Gustav Carus, Department of Dermatology, Skin Cancer Center at the University Cancer Centre and National Center for Tumor Diseases, TU Dresden, 01307 Dresden, Germany; Friedegund.Meier@uniklinikum-dresden.de

**Keywords:** melanoma, metastatic malignant melanoma, targeted therapy, immune checkpoint inhibitor, relative survival, cancer registry data

## Abstract

Targeted therapies (TT) and immune checkpoint inhibitors (ICI) have become increasingly important in the treatment of metastatic malignant melanoma in recent years. We examined implementation and effectiveness of these new therapies over time in Germany with a focus on regional differences. We analyzed data from 12 clinical cancer registries in 8 federal states in Germany over the period 2000–2016. A total of 3871 patients with malignant melanoma in Union internationale contre le cancer (UICC) stage IV at primary diagnosis (synchronous metastases) or with metachronous metastases were included. We investigated differences in survival of patients treated with new and conventional therapies by log-rank tests for Kaplan–Meier curves. Cox regression models were estimated to adjust therapy effects for demographic, regional, and prognostic factors. New systemic therapies were increasingly applied throughout Germany. TT were most frequently documented in Eastern Germany (East: 11.2%; West: 6.3%), whereas ICI therapies were more frequently used in Western Germany (East: 1.7%; West: 3.9%). TT had a relevant influence on patient survival (hazard ratio (HR) = 0.831; 95%-CI = (0.729; 0.948)). Survival was worse in Eastern Germany (HR = 1.470; 95%-CI = (1.347; 1.604)) relative to Western Germany. Treatment and survival prospects of patients with melanoma differed considerably between Western and Eastern Germany. The differences in regional medication behavior and survival require further exploration.

## 1. Introduction

More than 80% of new malignant melanoma cases worldwide occur in Australia, New Zealand, North America, and Europe [1]. It is the third most common cancer in Australia (age-standardized incidence rate 33.6 per 100,000, World Standard Population) and New Zealand (33.3 per 100,000), and the seventh most common cancer in North America (12.6 per 100,000) as well as in Europe (11.2 per 100,000). In Germany, malignant melanoma is the sixth most common cancer (21.6 per 100,000) [1]. The absolute number of incident melanoma cases increased continuously between 1999 and 2017 [2]. The age-standardized incidence rate increased significantly only in 2007–2008 and remained stable over the rest of the period, while the age-standardized mortality rate remained unchanged over the whole period. Malignant melanoma can be treated with an exceptionally good prognosis if it is detected in the early stages (Union internationale contre le cancer (UICC) 0-I) [3]. In contrast, patients with metastatic melanoma have a notably unfavorable prognosis for mortality. Up until 2008, patients in stage IV were mainly treated with chemotherapy (mostly cytostatics), which was also recommended in guidelines [4,5]. This therapy had only a limited effect on survival. Until recently, patients with resected high-risk melanoma and/or locoregional metastases were treated with the cytokine interferon. In a meta-analysis [6], the median overall survival for interferon compared with no interferon was 5.0 years versus 4.4 years; the overall survival hazard ratio (HR) was 0.91 (95%-CI = (0.85; 0.97)).

However, after decades of stagnation, recent advances in immunotherapy with immune checkpoint inhibitors (ICI) and targeted therapies (TT) with BRAF (v-raf murine sarcoma viral oncogene homolog B1) and MEK (mitogen-activated protein kinase kinase) inhibitors have considerably improved the prognosis of metastatic melanoma.

The serine/threonine kinase BRAF and the protein kinase MEK are members of the MAP (mitogen-activated protein) kinase signaling pathway. Activating mutations of BRAF result in uncontrolled tumor growth. The BRAFV600E mutation is present in approximately 40–60% of melanomas [7].

In Germany, the BRAF inhibitors vemurafenib and dabrafenib have been licensed for systemic treatment of melanoma since 2012 and 2013, respectively. In a phase 3 study, 675 treatment-naive patients with metastatic melanoma and BRAFV600E mutation were treated with vemurafenib or dacarbazine [8]. Median overall survival was 13.6 months for vemurafenib and 9.7 months for dacarbazine [9]. Dabrafenib showed comparable results in a phase 3 study in 250 patients [10].

Altogether, TT with BRAF inhibitors and MEK inhibitors provides rapid disease control with high response rates in patients with BRAFV600E-mutated metastatic melanoma. However, most patients develop resistance to therapy during the course of therapy.

Immunotherapies work by directing the attention of the immune system against the cancer to actively kill the tumor cells [11]. Peptides derived from tumor-associated antigens are presented by the major histocompatibility complex on the surface of dendritic cells and recognized by T cells via their T cell receptor. The co-stimulatory molecules B7-1 and B7-2 are required for T cell priming. T cell activation leads to upregulation of CTLA-4 (cytotoxic T lymphocyte-associated antigen 4) on T cells. Binding of CTLA-4 to B7 receptors of dendritic cells results in inhibition of T cell activation. Anti-CTLA-4 antibodies restore T cell stimulation in the lymph nodes. Following long-term stimulation, the PD-1 (programmed death 1) receptor is upregulated by T cells. Its ligand PD-L1 is expressed on cancer cells and binds to PD-1 receptors on T cells, which leads to their inhibition. Anti-PD-1/PD-L1 antibodies enhance the functional properties of effector T cells at the tumor site.

The CTLA-4 antibody ipilimumab was the first systemic therapeutic agent to achieve a significant prolongation of overall survival in patients with metastatic melanoma [12]. In a phase 3 study, previously treated patients with metastatic melanoma were treated with ipilimumab or with a gp100 peptide vaccine (gp100) or with ipilimumab plus gp100. Median overall survival was 10 months for ipilimumab, 6 months for gp100, and 10 months for ipilimumab plus gp100.

Nivolumab was the first anti-PD-1 antibody approved for the treatment of melanoma by the European Medicines Agency in 2015. In a phase 3 study [13], 418 previously untreated patients with metastatic melanoma and wild-type BRAF received either nivolumab or dacarbazine. Nivolumab compared with dacarbazine achieved objective response rates of 40.0% vs. 13.9%. Median overall survival in nivolumab-treated patients (62% with two to five prior systemic therapies) was 16.8 months [14]. Pembrolizumab is another anti-PD-1 antibody approved by the European Medicines Agency in 2015. The studies conducted to date showed comparable results in terms of both efficacy and toxicity [15].

Compared to TT, ICI appear to act slower and achieve lower response rates. However, ICI achieve durable responses. One strategy currently under investigation is combining BRAF/MEK inhibitors with ICI. This combination strategy combines the hope for a fast, reliable, and lasting response to therapy. Preclinical and early clinical data support this hypothesis [16,17,18,19].

However, in a comprehensive literature search, we did not find any publications on the implementation of promising new systemic therapies, particularly TT and ICI. Against this background, an assessment of their use in Germany based on cancer registry data is of high interest and contributes to fill this research gap. The first objective of our analysis therefore was to describe the frequency of systemic therapies used in the treatment of metastatic malignant melanoma in the period from 2000–2016 in Germany. This 17-year observation period covers times before and after the approval of first drugs regarding TT and ICI.

The second objective of the analysis was to assess the differences in overall survival regarding innovative TT and ICI compared to chemotherapy and interferon therapy. Furthermore, we estimated survival effects of the applied therapies, adjusting for anatomic site (localization) and melanoma subtypes (morphology) of the primary tumor, synchronous vs. metachronous metastasis, and sociodemographic variables. The robustness of the findings was assessed by sensitivity analyses.

Regional differences in cancer diagnoses and mortality in Germany were reported and discussed for skin cancer [20,21,22,23,24,25], lung cancer [26], and colorectal cancer [27]. Because of these reported differences and our own clinical experience at the Skin Cancer Center at the University Cancer Centre Dresden, we paid attention to potential differences between Western and Eastern Germany regarding the first and second objective.

## 2. Results

### 2.1. Baseline Characteristics

The study cohort comprised 3871 patients, 1080 of whom were residents of the West German federal states and 2791 of the East German federal states (Table 1). The median age at metastasis was 67 years (Q1 = 55; Q3 = 75), with similar results for Eastern and Western Germany. With a higher share of males in each cohort, the sex distribution was also similar between regions. The occurrence of melanoma subtypes was reported in nearly the same order in both subgroups, but nodular melanoma (NM) were documented considerably more often in the East German federal states (32.6%, 95%-CI = (30.8%; 34.3%)) than in the West German states (21.2%, 95%-CI = (18.8%; 23.8%)). Differences were found for metastasis type and anatomic sites.

Regarding therapies, the descriptive analysis revealed that interferon therapy was the most commonly applied systemic therapy in both parts of Germany, with a higher share of 20.0% (95%-CI = (18.5%; 21.5%)) in Eastern Germany compared to Western Germany with a proportion of 14.5% (95%-CI = (12.5%; 16.8%)). Targeted therapy was more frequently applied in the Eastern states (11.2%, 95%-CI = (10.0%; 12.4%)) than in the Western states (6.3%, 95%-CI = (4.9%; 7.9%)). An inverse relation was found for ICI, which was applied in 3.9% (95%-CI = (2.8%; 5.2%)) of the patients in the West, and 1.7% (95%-CI = (1.3%; 2.3%)) in the East.

### 2.2. Application of ICI and TT in Germany

All over Germany, the application of TT increased over the observation period, especially starting in 2008. In Western Germany, TT have been applied at a lower level compared to Eastern Germany (Figure 1).

The treatment with ICI, however, has developed differently relative to application of TT. The figure indicates annual increases since 2012.

Until now, combined ICI and TT has been used less often than the other considered therapies. According to the data, combined TT and ICI was applied more frequently in Western than in Eastern Germany.

### 2.3. Relative Survival

Metastasized melanoma patients generally had a poor survival prospects (relative 5-year survival rate: 21.9%, 95%-CI = (20.3; 23.5)). However, stratification by therapy type, metastasis type, and region individually revealed substantial differences (Figure 2). The differences were statistically insignificant for chemotherapy and interferon therapy, but significant for all other strata (ICI, TT, metastasis type, region).

While relative survival analysis relates observed mortality of metastasized melanoma patients to the mortality of the total German population, it does not completely adjust survival for patient-specific covariates. Such adjustment was performed using multivariable Cox regression analysis, as shown in the following section.

### 2.4. Cox Regression Analysis

Cox regression analyses were conducted for all 3871 patients with an overall survival of at least 1 day since metastasis (Table 2 and Appendix A). Table 2 shows results of univariable Cox regression analysis (column 1), multivariable Cox regression analysis (column 2), and an extended multivariable Cox regression analysis with time-dependent coefficients (column 3). The latter relaxes the proportional hazards assumption of the Cox model and allows for time-varying therapy effects.

The multvariable Cox proportional hazards model with only time-constant effects (see column 2) suggested statistically significant protective effects of targeted therapy (HR = 0.748; 95%-CI = (0.657; 0.853)) and interferon therapy (HR = 0.833; 95%-CI = (0.754; 0.919)). Chemotherapy was related to a significantly higher hazard rate (HR = 1.172; 95%-CI = (1.038; 1.323)). The effect of ICI was insignificant (see column 2). However, all therapies and the metastasis types violated the proportional hazards assumption, which may indicate time-varying effects.

Therefore, for all variables that violated the assumption of proportional hazards, an additional time-dependent effect was estimated (column 3). In this model, the effect of therapies and metastasis types on overall survival was represented by two coefficients for each covariate: a time-constant coefficient and a time-varying coefficient. The latter was calculated from the interaction of the respective covariate with (log.) time since metastasis and thus allowed for estimation of time-varying effects of covariates on the patients’ survival.

While the time-constant coefficient of targeted therapy indicated a hazard ratio below unity (HR = 0.831; 95%-CI = (0.729; 0.948)), the time-dependent coefficient of targeted therapy showed a hazard ratio above unity (HR = 1.585; 95%-CI = (1.412; 1.779)). In combination, these coefficients may be interpreted as follows: At the beginning of therapy, there was a protective effect of TT. With an interaction with time, i.e., as time progresses, the protective effect diminished.

The same pattern was found for interferon therapy. A protective effect early after metastasis was indicated by a statistically significant time-constant coefficient (HR = 0.899; 95%-CI = (0.812; 0.995)). Over time, this protective effect on survival became smaller (in absolute terms) as indicated by the time-dependent coefficient of interferon therapy (HR = 1.246, 95%-CI = (1.154; 1.346)).

In patients treated with chemotherapy, the results showed statistically significant time-constant and time-dependent effects, both implying an increase in patients’ hazard rate.

The time-constant coefficient of ICI was not significant. However, the time-dependent coefficient of ICI showed a significant increase in hazard over time (HR = 1.363; 95%-CI = (1.067; 1.742)).

To visualize estimated differences between therapies, we estimated survivor functions for each therapy type on the basis of their respective time-constant and time-dependent coefficients (Figure 3). As already indicated by the results in Table 2, the predicted survivor curves showed the largest protective effect for TT, followed by interferon therapy. For both therapies, there was a decrease in protective effects over time. A similar pattern was observed for ICI. For patients treated with chemotherapy, the survivor curve showed the earliest intersection with the survival curve for patients without documented therapy.

Further substantial associations with survival were found for metastasis type. This holds for both the univariable (HR=1.413, 95%-CI = (1.309; 1.526)) and the multivariable (HR = 1.281, 95%-CI = (1.171; 1.401)) model. Patients with metachronous metastasis had a 28% higher hazard of death than patients with synchronous metastasis. Additionally, including a time interaction revealed a significant time-dependent effect of metachronous metastasis showing that differences between metastasis types decreased over time since metastasis.

All socio-demographic variables showed statistically significant effects. Mortality risk increased by 1.1% with every year of age (HR = 1.011; 95%-CI = (1.009; 1.014)). Another effect was observed for the patient’s sex, implying better survival prospects for women than for men (HR = 0.869; 95%-CI = (0.806; 0.938)). Furthermore, patients from Eastern Germany had a 47% higher hazard of death compared to patients from Western Germany (HR = 1.470; 95%-CI = (1.347; 1.604)).

In addition to the coefficients reported here, the survival analysis also adjusted for localization and melanoma subtype of the primary tumor. These results are shown in Appendix A. Coefficients for melanoma subtypes were not significant.

Sensitivity tests were conducted (a) to avoid biases due to documentation effects and behavior of different registries and treating physicians, (b) to differentiate therapy effects and side effects (to avoid distortions due to inclusion of patients who were not receiving therapy because of poor survival prospects at diagnosis), (c) to avoid distortions due to the application of TT and ICI as adjuvant therapies, (d) to observe possible effects of skin cancer screening, and (e) to identify distortions in treatment effects due to regional characteristics or metastasis types.

Sensitivity tests used different subsets of the dataset (a,b,c), an additional covariate (d), or stratification by region and type of metastasis (e). None of the sensitivity analyses regarding (a–d) induced qualitative changes in the multivariable regression coefficients (Appendix A). Stratification by region and metastatic type (e) revealed evidence for effect modifications in some covariates (Appendix A)—the effects of anatomic site and type of metastasis on survival differed between Eastern and Western Germany. Furthermore, we found different effects of anatomic site and interferon therapy in patients with metachronous and synchronous metastases, respectively. Although estimated effects differed quantitatively, the direction of the effect regarding the covariate type of metastasis was the same in the considered subgroups.

A protective effect of interferon therapy was revealed only in the subgroup of synchronously metastasized patients. This effect was also observed in the main analysis of all patients.

## 3. Discussion

This study investigated the application of systemic therapies in patients with metastasized malignant melanoma. The analyzed study population is a small proportion of all melanoma patients and therefore can be compared to other studies only to a limited extent.

In 2015–2016 in Germany, patients with a stage UICC IV melanoma at primary diagnosis had a share of 4% in women and 5% in men in all diagnosed malignant melanoma [2]. In a long-term analysis of German cancer registry data between 2002–2011 [25], stage UICC IV accounted for 2.2% of all primary diagnoses. While the proportion of women and men in all diagnosed melanoma patients in 2016 in Germany was nearly balanced (women: 48.0%; men: 52.0%) [2], our sample of patients with advanced melanoma was characterized by a higher share of men (women: 39.5%; men: 60.5%). These results are in line with the findings reported in [25], where the proportion of women and men in stage UICC IV at primary diagnosis was 39.4% and 60.6%, respectively.

In terms of age distribution, patients with stage UICC IV at primary diagnosis slightly differed from all melanoma patients. A smaller proportion of 3.5% was reported in the age group of 15–34 (vs. 7.9% in all melanoma patients) and a proportion of 15.9% in the age group of 35–49 (vs. 19.2% in all melanoma patients). Starting at the age of 50 years, the share of patients with primary tumor at stage UICC IV was about 3% higher in each age group compared to all melanoma patients [25].

According to our first objective, in the analyzed data the application of TT and ICI increased since 2008 and 2012, respectively. TT was more frequently documented in Eastern Germany. Regional differences could be due to different healthcare provider structures, differences in localization and severity of metastases, and differences in reporting behavior of the treating physicians that may cause documentation biases.

In our data, ICI was the least applied therapy in both parts of Germany. This was mainly caused by the later approval of anti-PD-1 antibodies compared to BRAF inhibitors. Therefore, the number of patients treated by ICI was small in the analyzed sample.

To address our second objective, we inspected survival regarding several therapies. While unadjusted relative survival curves did not differ significantly regarding chemotherapy and interferon therapy, ICI and TT were associated with substantially higher survival rates over a period of five years since metastasis.

Altogether, the effect of TT and ICI on survival prospects compared to the effect of chemotherapy is in line with general findings for the BRAF inhibitors vemurafenib and dabrafenib vs. dacarbazine [9,10], and anti-PD-1 antibody nivolumab vs. dacarbazine [13].

Huge differences in survival were further shown for metastasis type and region, indicating significantly higher mortality rates in metachronously than in synchronously metastasized patients and in patients from Eastern compared to patients from Western Germany.

In line with these results, after adjusting for covariates, Cox regression results indicated that TT had the largest protective effect on survival regarding therapies. We also found evidence for protective effects of TT, which decreased over time since metastasis. Regarding ICI, we found a similar pattern. However, the protective effect was not significant, most likely due to the small number of cases.

In contrast to the results from relative survival rates discussed above, Cox regression indicated a protective effect of interferon therapy on survival that diminishes over time. However, subgroup analysis revealed this effect only in synchronously metastasized patients. There was no evidence for a positive impact of chemotherapy on survival.

The results on metastasis type indicated a higher mortality risk in patients with metachronous metastases compared with patients with synchronous metastases. The difference between metachronous and synchronous metastases in terms of survival decreased over time. One of the largest effects was estimated for the patients’ place of residence, indicating that the mortality risk in patients from Eastern Germany was 47% higher compared to patients from Western Germany. This finding is in line with previous studies on differences between Western and Eastern Germany, which indicated less favorable diagnoses and prognoses for patients with malignant melanoma in Eastern Germany [20,21,22,23,24,25]. As reasons for the regional differences, variations in the quality of care including the number of dermatologists, documentation, demographic and socio-economic population structure, travel behavior, and UV radiation were discussed [22,23].

No qualitative changes were observed in the regression results when accounting for possible documentation effects, side effects, beginning of documented therapy, or effects of skin cancer screening.

Effects regarding the covariates of anatomic site differed between Eastern and Western Germany and between metachronous and synchronous metastases. The effect regarding the covariate type of metastasis was stronger in patients from Western Germany than from Eastern Germany but had the same direction. However, there was no evidence for differences in effects of chemotherapy, TT, and ICI between the subgroups. Group differences with regard to region and type of metastasis are quite relevant in the care of melanoma patients. However, they do not affect the interpretation of the results on effects of chemotherapy, TT, and ICI, whose estimation and comparison was the second objective of this study.

Further research regarding systemic therapies in the treatment of metastatic malignant melanoma should analyze a larger observation period to cover a sufficient number of cases of innovative therapies and combined therapies. Statutory health insurance data could be used to improve data quality, especially in terms of therapy detection and therapy beginning. A further progress would be the inclusion of region-specific data to better reflect regional differences.

A limitation of the study is that TT and ICI have been implemented mainly in patients metastasized since 2008, with the observation period ranging from 2000 to 2016. For this reason, fewer cases of TT and ICI than chemotherapy and interferon therapy were documented in the data. At the same time, the possible follow-up times in patients treated with TT and ICI were more limited than in patients treated with chemotherapy or interferon therapy. In particular, follow-up after approval of combined therapies with TT and ICI was very short in the analyzed data. Therefore, it was not included in the analysis.

Another limitation of our study relates to the quality of the obtained registry data. Some desirable information, e.g., localization of metastases, was not recorded or has not been available for analysis. Other variables such as center treatment or early cancer detection were handled differently by different registries, which resulted in missing values. This was because registries in Germany are obliged to collect data that are specified in a uniform oncological baseline dataset (Einheitlicher onkologischer Basisdatensatz), in which the latter variables are not included. Documentation differences between registries could further result from differences in available resources, oncological certifications, and state laws.

Our data included a smaller number of patients from Western Germany compared with Eastern Germany. Findings for West German patients therefore may be less generalizable than for patients from Eastern Germany. Nevertheless, both groups contained a sufficiently high number of patients to conduct survival analysis.

For the majority of considered patients, no systemic therapy or treatment was documented. This may have resulted in a loss of evidence in the survival analysis if the reference group may contain patients who were erroneously considered to have not received therapy.

Regarding therapy information, it is important to note that the treating physician only reported to a registry on specific occasions such as diagnosis and start of therapy. An explicit statement that no therapy was performed or that a specific therapy was not performed therefore cannot be systematically derived from the data. In the absence of therapy information, it was therefore not possible to distinguish between therapy that was not carried out and therapy that was carried out but not documented.

The risk of confounding by indication cannot be completely excluded from the analysis. Adjusting for covariates and socio-demographics was carried out to avoid such effects as much as possible.

The strengths of the study result primarily from data characteristics. Application and treatment effects of innovative TT and ICI were assessed on a broad basis over a period of 17 years. By drawing on data from different regions in Germany, a relatively high number of patients with metastatic melanoma could be included. Furthermore, due to validation of death notices by most registries, the cancer registry data are a valid and reliable source regarding incidence of malignant melanoma, treatment and medication, clinical information, and patient’s life status.

## 4. Materials and Methods

### 4.1. Data Collection

In June 2017, clinical and epidemiological cancer registries in Germany were contacted by the Working Group of German Tumor Centers and Clinical Cancer Registries (ADT) and were asked to provide anonymized data on malignant melanoma patients diagnosed between 2000 and 2016 for analysis for the Federal Oncological Quality Conference 2018 [28]. In addition, 12 of these clinical cancer registries agreed by September 2019 to use these data for the present analysis with the purpose of describing ICI and TT in Germany. Thus, data from registries in the federal states of Bavaria, Baden-Württemberg, North Rhine-Westphalia, Hesse, Berlin, Saxony, Saxony-Anhalt, and Thuringia could be included. Some registers also included patients living outside their own state. Thus, data were available from patients with place of residence in all federal states, except Saarland.

The data provided information on sex, place of residence, date of birth, life status, date of primary diagnosis, date of diagnosis of distant metastasis, last contact date, date of death, applied therapy, substances of systemic therapy, beginning of therapy (except TT), and anatomic site (localization) and subtype (morphology) of primary tumor. Further available information (general performance status, presence of BRAF/MEK mutation, center treatment, lymph node surgery, surgery resection, participation in early detection of cancer programs) could not be considered in the analysis because of a high proportion of missing values or insufficient variance. Prognostic factors, such as the beginning of targeted therapy or the localization and morphology of distant metastasis, were not available.

### 4.2. Definition of Study Populations

In total, the cancer registries requested submitted data on 46,160 patients with malignant melanoma with a primary diagnosis between 2000–2016. Patients for whom no metastases were documented were excluded from the analysis.

Study populations were then defined using the date of metastasis and the patient’s overall survival since metastasis. In further steps, all patients were excluded for whom metastases were identified outside the 2000–2016 observation period or who died at the same day that the diagnosis of distant metastases was made. The remaining patients were included in descriptive and survival analyses.

Patients who deceased within 30 days after metastasis further reduced the study population. This excluded the patients for whom it could not be clarified whether they had received therapy before death, or whether they could benefit from therapy. The remaining group was used to conduct sensitivity tests for survival analysis. All steps in defining study populations and the remaining sample sizes are shown in Figure 4.

### 4.3. Data Transformation

Overall survival was defined as the survival time since metastasis. The metastasis date was documented in the original data as the date on which distant metastases were identified if a primary tumor had previously been diagnosed. In patients with a stage UICC IV, the date of metastasis was equivalent to the diagnosis date of the primary tumor.

Overall survival was calculated as the time span from metastasis to death in patients who died during the 2000–2016 observation period. In patients who did not die, survival was calculated from the date of metastasis and 31 December 2016 for registries that regularly synchronize data with death notices and registration offices. In patients alive for whom the registries did not carry out such comparison, we set the end date for survival to the last known date on which the patient was alive. Patients who did not die during the observation period were treated as censored.

The original registry data differentiated between chemotherapy, immunotherapy (with no further differentiation between interferon therapy and ICI), and TT. The therapy received by a patient was not consistently documented in the data. It was therefore collated and corrected using information on documented substances of systemic therapy. Substances of systemic therapy were extracted from the dataset and assigned to four types of therapy: interferon therapy, chemotherapy (cytostatic), immune checkpoint inhibitor therapy (monoclonal antibodies), and targeted therapy (BRAF and MEK inhibitors). A dichotomous variable indicating whether or not the respective therapy was applied to a patient was created for chemotherapy, interferon therapy, ICI, and TT, individually. It should be noted that ICI could be identified only by the reported medication. Since both interferon therapy and ICI were documented by the registries in a single variable “immunotherapy”, we assumed a patient documented as treated with “immunotherapy” and for whom no medication was specified to have had received interferon therapy. That may explain why the number of cases with ICI treatment in the sample was comparatively small.

Information on synchronous and metachronous metastasis was derived from the difference between metastasis date and diagnosis date. Synchronous metastasis was defined as distant metastases occurring within 3 months after the primary diagnosis. Metastases occurring beyond this period were considered as metachronous.

The anatomic site (localization) and the melanoma subtype of the primary tumor (histology/morphology) were recorded in the registry data in a differentiated manner. For the purpose of the analysis, we grouped anatomic sites into melanoma of head and neck (ICD-10 C43.0–43.4), melanoma of trunk (ICD-10 C43.5), melanoma of extremities (ICD-10 C43.6 and C43.7), and melanoma of overlapping and unspecified sites of skin (ICD-10 C43.8 and C43.9).

Melanoma subtypes were categorized into malignant melanoma, not otherwise specified (MM NOS; ICD-O-3 8720/3); nodular melanoma (NM; ICD-O-3 8721/3); lentigo maligna melanoma (LMM; ICD-O-3 8742/3), superficial spreading melanoma (SSM; ICD-O-3 8743/3), and acral lentiginous melanoma (ALM; ICD-O-3 8744/3). Other subtypes documented in the registry data were assigned to a category “other”.

The patient’s age at metastasis was calculated as the difference in years between the metastasis date and the date of birth.

The region where a patient lived was derived from the postal zip code of the place of residence as reported in the registry data. Zip codes were recoded into the German community identification number (Amtlicher Gemeindeschlüssel, AGS) that contains information on the federal state in which a municipality is located. The states were then grouped into the federal states of Western Germany and the federal states of Eastern Germany including Berlin (the following federal states were assigned to Western Germany: Baden-Württemberg, Bavaria, Bremen, Hamburg, Hesse, Lower Saxony, North Rhine-Westphalia, Rhineland-Palatinate, Saarland, Schleswig-Holstein; the following federal states were assigned to Eastern Germany: Brandenburg, Mecklenburg-West Pomerania, Saxony, Saxony-Anhalt, Thuringia, Berlin).

Additionally, a dichotomous variable was derived from the date of the initial diagnosis, which indicates whether the diagnosis was made before 1 July 2008. This variable was used for sensitivity analyses and represents the (date of the) nationwide introduction of the statutory skin cancer screening.

To assess the application of ICI and TT over time in the western and eastern federal German states, we considered TT as having begun in the year of metastasis. For ICI, a start time was available and used to describe the application, unless the date was missing. In the latter cases, the year of metastasis was also used.

### 4.4. Statistical Analysis

Kaplan–Meier curves were used to compare the effects of therapy, region, and synchronous vs. metachronous metastases on overall relative survival over a period of 5 years since metastasis. Relative survival was estimated according Ederer II [29,30]. Nonparametric log-rank tests were used to evaluate differences between the curves. Period life tables were obtained from the Human Mortality Database [31].

A Cox proportional hazards model was used to estimate therapy effects on overall survival adjusted for covariates. The following variables were included in the model: sex, age at diagnosis of metastasis, region in which the patient lived (Western Germany; Eastern Germany), anatomic site of the primary tumor (unspecified/overlapping; head/neck; trunk; extremities), melanoma subtype of the primary tumor (other; MM NOS; NM; LMM; SSM; ALM), metastasis type (synchronous; metachronous), chemotherapy (no; yes), interferon therapy (no; yes), ICI (no; yes), TT (no; yes).

Variables were tested for the proportional hazards assumption using chi-square tests based on [32]. Those who violated the assumption were included in the model with additional time-varying coefficients, on the basis of an interaction of the respective variable with the log-transformed time since metastasis. Variables that indicate combined treatments (e.g., interaction of TT and ICI) were not included in the model due to a low number of cases. A *p*-value of 0.05 or lower was considered to indicate significance.

Findings of Cox regression analysis were used to predict survivor functions for each therapy type on the basis of average values of the remaining covariates.

### 4.5. Sensitivity Analyses

A first sensitivity analysis was carried out to avoid documentation biases due to the missing documentation of therapies or documentation differences over time. The Cox regression analysis was therefore carried out for five subsets of the entire sample. For each subset, a cutoff value was defined, which limited the share of patients per registry and year for whom no therapy was documented. This share ranged from 50% to 90%.

A second sensitivity analysis was carried out to differentiate therapy effects from side effects. Since the start of therapy was not documented in the registry data for each therapy, it was difficult to determine whether early deceased patients could benefit from systemic therapy. The Cox regression model described above was therefore estimated only for patients with an overall survival of at least 31 days since metastasis.

A third sensitivity analysis was carried out to avoid distortions due to application of TT and ICI as adjuvant therapies. The Cox regression analysis was therefore conducted using a reduced sample with cases from 2012 and thereafter only. In addition, patients treated with ICI before 2015 were excluded. Regarding TT an ICI, the reduced sample therefore consisted of patients who were treated with TT and ICI from the time the therapies were approved as first-line therapies in Germany.

A fourth sensitivity analysis was carried out to observe possible effects of skin cancer detection. For this purpose, the regression model was estimated including a dummy variable capturing whether primary diagnosis was made before July 2008. At this time, skin cancer screening became a health insurance benefit in Germany for patients at the age of 35+.

A last sensitivity analysis was carried out to explore possible differences in effects between subgroups. Therefore, the Cox regression model was estimated, being stratified by region and by type of metastasis.

For the first four sensitivity analyses, we assessed whether they caused a qualitative change in the Cox regression coefficients. A qualitative change was defined as a change in the sign of the coefficient without loss of statistical significance. In the fifth sensitivity analysis, we assessed whether effect modifications occurred. An effect modification was assumed if the 95%-CI of estimated effects of a specific covariate in different subgroups did not overlap.

### 4.6. Software

All statistical analyses were performed using R version 3.6.1 [33]. Survival analyses were performed using the additional R packages “survival” [34,35] and “survminer” [36]. Survival parameter estimates were computed using the function coxph and its time-transform functionality contained in the “survival” package. Relative survival was estimated using the R package “relsurv” [37].

## 5. Implications

Our analysis provided relevant insights on the application and treatment effects of systemic therapies in Germany during 2000–2016. New systemic therapies were increasingly applied throughout Germany.

Treatment and survival prospects of patients with metastatic melanoma differ considerably between Western and Eastern Germany. Identifying the underlying causes of regional differences in adherence to guidelines including recommended systemic treatment will require more in-depth studies. There may be several barriers to guideline adherence including lack of familiarity with guidelines, lack of agreement in the benefits of treatment versus the risks, inertia of previous practice, and patient-related barriers. Future studies of adherence to recommended treatment should be designed to examine all of these possibilities to ensure a guideline-based and high-quality treatment of all patients with melanoma.

Furthermore, an updated analysis of the application and treatment effects of systemic therapies in Germany during 2017–2020 is of high relevance because new systemic therapies for patients with metastatic melanoma were approved, in particular the immune checkpoint inhibitor combination nivolumab plus ipilimumab that was approved in 2016 and was reported to achieve a 5-year survival rate of more than 50% in 2019 [38].

## Figures and Tables

**Figure 1 cancers-12-02354-f001:**
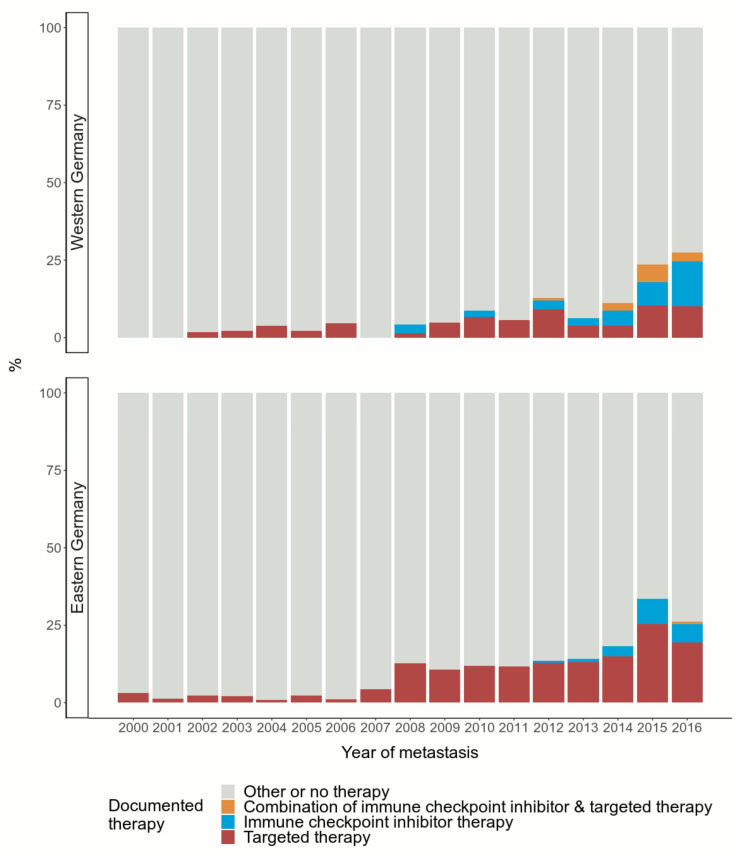
Documented application of immune checkpoint inhibitor therapy and targeted therapy over time during the 2000–2016 observation period stratified by region. Number of patients: *n* = 3871 (*n* = 1080 in Western Germany; *n* = 2791 in Eastern Germany).

**Figure 2 cancers-12-02354-f002:**
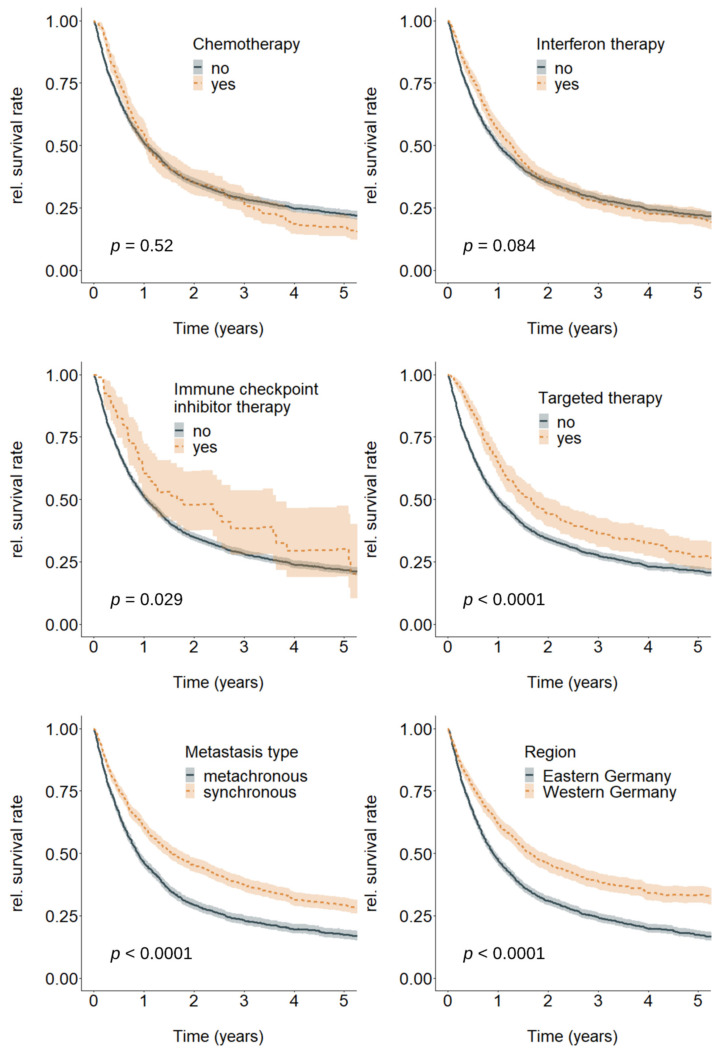
Overall relative survival by chemotherapy (no; yes), interferon therapy (no; yes), immune checkpoint inhibitor therapy (no; yes), targeted therapy (no; yes), metastasis type (metachronous; synchronous), and region (Eastern Germany, Western Germany), *n* = 3871. *p*-value shows the results of Log rank tests for differences between the respective strata.

**Figure 3 cancers-12-02354-f003:**
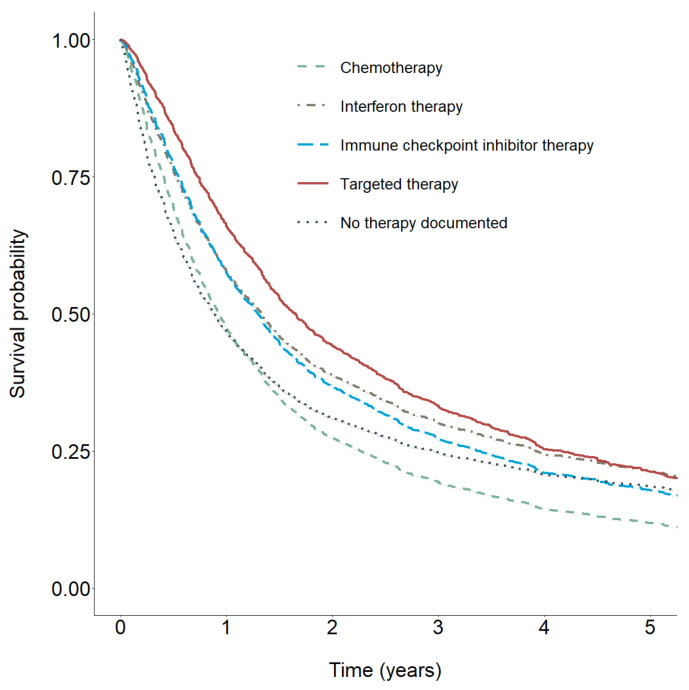
Predicted survivor functions (mean adjusted) for chemotherapy, interferon therapy, immune checkpoint inhibitor therapy, targeted therapy, and no documented therapy.

**Figure 4 cancers-12-02354-f004:**
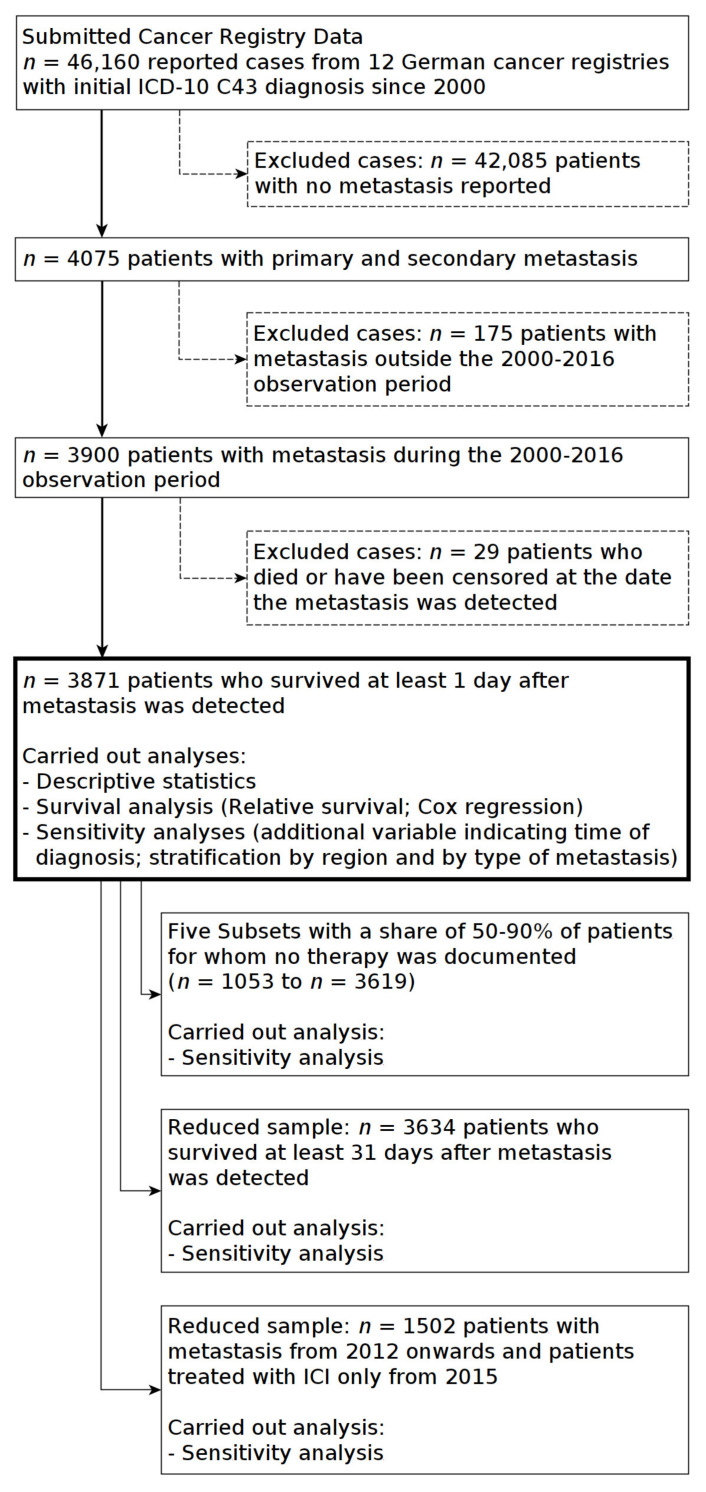
Selection of study populations from cancer registry data.

**Table 1 cancers-12-02354-t001:** Characteristics of the study cohort and important subgroups.

Region	Western Germany		Eastern Germany		Overall Sample	
*n*	1080		2791		3871	
Characteristic	Median/*n*	Q1; Q3/%	Median/*n*	Q1; Q3/%	Median/*n*	Q1; Q3/%
Age at metastasis, median (Q1; Q3)	65	(53; 74)	67	(55; 76)	67	(55; 75)
Sex						
Female, *n* (%)	443	(41.0)	1085	(38.9)	1528	(39.5)
Male, *n* (%)	637	(59.0)	1706	(61.1)	2343	(60.5)
Metastasis type						
Synchronous metastasis, *n* (%)	569	(52.7)	783	(28.1)	1352	(34.9)
Metachronous metastasis, *n* (%)	511	(47.3)	2008	(71.9)	2519	(65.1)
Anatomic site of primary tumor (localization)						
Head/neck, *n* (%)	157	(14.5)	453	(16.2)	610	(15.8)
Trunk, *n* (%)	293	(27.1)	946	(33.9)	1239	(32.0)
Extremities, *n* (%)	398	(36.9)	1205	(43.2)	1603	(41.4)
Unspecified/overlapping, *n* (%)	232	(21.5)	187	(6.7)	419	(10.8)
Melanoma subtype of primary tumor (morphology)						
Malignant melanoma, NOS, *n* (%)	499	(46.2)	1113	(39.9)	1612	(41.6)
Nodular melanoma, *n* (%)	229	(21.2)	909	(32.6)	1138	(29.4)
Lentigo maligna melanoma, *n* (%)	19	(1.8)	87	(3.1)	106	(2.7)
Superficial spreading melanoma, *n* (%)	200	(18.5)	446	(16.0)	646	(16.7)
Acral lentiginous melanoma, *n* (%)	44	(4.1)	118	(4.2)	162	(4.2)
Other, *n* (%)	89	(8.2)	118	(4.2)	207	(5.3)
Documented therapy						
Chemotherapy, *n* (%)	107	(9.9)	242	(8.7)	349	(9.0)
Interferon therapy, *n* (%)	157	(14.5)	557	(20.0)	714	(18.4)
Immune checkpoint inhibitor therapy, *n* (%)	42	(3.9)	48	(1.7)	90	(2.3)
Targeted therapy, *n* (%)	68	(6.3)	312	(11.2)	380	(9.8)

*n* = number of observations, Q1 = first quartile, Q3 = third quartile, NOS = not otherwise specified. Percentages refer to the number of observations in the groups.

**Table 2 cancers-12-02354-t002:** Univariable and multivariable Cox regression results for survival since diagnosis of first metastasis, adjusted for age at metastasis, sex, region, anatomic site of the primary tumor, melanoma subtype of the primary tumor, metastasis type, and therapy (*n* = 3871; events: 2965).

Characteristic ^1^	Hazard Ratio (95%-CI) (Univariable)	Hazard Ratio (95%-CI) (Multivariable, Time-Constant Effects)	Hazard Ratio (95%-CI) (Multivariable, Time-Constant, and Time-Dependent Effects)
Age at metastasis (years)	1.014 (1.011; 1.017)	1.012 (1.009; 1.015)	1.011 (1.009; 1.014)
Sex: female (ref: male)	0.846 (0.786; 0.911)	0.867 (0.803; 0.935)	0.869 (0.806; 0.938)
Region: Eastern Germany (ref: Western Germany)	1.517 (1.395; 1.650)	1.456 (1.334; 1.589)	1.470 (1.347; 1.604)
Metachronous metastasis (ref: no metachronous metastasis)	1.413 (1.309; 1.526)	1.281 (1.171; 1.401)	1.222 (1.113; 1.341)
Interaction metachronous metastasis with time since metastasis (years)	-	-	0.851 (0.795; 0.910)
Chemotherapy: yes (ref: no)	1.040 (0.924; 1.170)	1.172 (1.038; 1.323)	1.237 (1.094; 1.398)
Interaction chemotherapy: yes with time since metastasis (years)	-	-	1.266 (1.146; 1.398)
Interferon therapy: yes (ref: no)	0.922 (0.840; 1.011)	0.833 (0.754; 0.919)	0.899 (0.812; 0.995)
Interaction interferon therapy: yes with time since metastasis (years)	-	-	1.246 (1.154; 1.346)
Immune checkpoint inhibitor therapy: yes (ref: no)	0.738 (0.561; 0.970)	0.894 (0.676; 1.182)	0.980 (0.738; 1.301)
Interaction immune checkpoint inhibitor therapy: yes with time since metastasis (years)	-	-	1.363 (1.067; 1.742)
Targeted therapy: yes (ref: no)	0.762 (0.671; 0.867)	0.748 (0.657; 0.853)	0.831 (0.729; 0.948)
Interaction targeted therapy: yes with time since metastasis (years)	-	-	1.585 (1.412; 1.779)

^1^Table 2 shows selected results of the Cox regression analyses. Complete findings (including results for anatomic sites and melanoma subtypes) are displayed in Appendix A.

## Data Availability

Restrictions apply to the availability of the data analyzed in the study.

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
