# Peer review of "Targeted and Checkpoint Inhibitor Therapy of Metastatic Malignant Melanoma in Germany, 2000–2016"

_cancers, 2020, doi:10.3390/cancers12092354_

Round 1
Reviewer 1 Report
Although in my opinion there are still some weaknesses in the design and desription of the results, the authors have improved the manuscript that could be now accepted for publication, considering the interest of the topic.
Reviewer 2 Report
Thank you very much for addressing my comments. I accept your article in the present form.
Reviewer 3 Report
While I am still leery of the Kaplan-Meier graph of the authors regional analysis in Figure 2, they have sufficiently explained the potential risks of overinterpreting the regional differences in the body of the manuscript.
Reviewer 4 Report
The Authors aimed to provide a comprehensive analysis/ review about melanoma treatments and their effectiveness in the last two decades, with a special focus on comparing regional differences in Germany.
While the overall intention to review effectivness of melanoma treatments is valuable, one may have the feeling that no clear-cut conclusions are made, and the reader is lost in the details and analysis methodology.
Suggestions, comments to improve:
- Fig. 2 is suprising. Regional difference is highlighted, but we do not know much more on treatment effects. Bottom left figure uncovers that maybe patients, cases should be subgrouped for analysis to gain more clinical insight. It seems scientifically, clinically uninteresting if there is a regional difference. You may reanalyse the data accordingly. If regional effect masks treatment differences, the analysis may be done for E W Ger for subgroups. It would be interesting to see if treatment effects follow the same trend?
- Methodology should be separated for better readership. For presenting data meaning of the analysis (what the analysis provides) should be concisely described.
- Discussion is pointing to?
Author Response
Please see the attachment.

This manuscript is a resubmission of an earlier submission. The following is a list of the peer review reports and author responses from that submission.
Round 1
Reviewer 1 Report
In the manuscript submitted by Hellmund et al., the Authors aim to investigate the impact of conventional therapies (i.e. chemotherapy and interferon therapy) versus novel therapies (i.e. targeted therapy (TT) and immune checkpoint inhibitors (ICI)) on the survival of metastatic malignant melanoma patients. Authors focus their study on the German landscape in the period between 2000-2016, with a particular regard to the differences between Western and Eastern Germany.
Despite the topic is undoubtedly of interest for the field, several weaknesses could be highlighted in this work. One critical issue is the difficulty to follow the paper, highly statistical and in my opinion not completely suitable for a publication in a high-impact Journal such as Cancers. But the main concern in my opinion regards the robustness of the data, as it is also discussed. Indeed, the fact that for the majority of included patients, no systemic therapy or treatment was documented (as discussed by the same Authors), could led to severe consequences on the interpretation of the results, that in this way loose their robustness and their meaning. While Authors identify the strengths of the study in data characteristics.
Authors state that application and treatment effects were assessed on a broad basis over a period of 17 years, covering the times before and after the approval of TT and ICI, however some discrepancies could be underlined. In the Introduction, Authors indicate that TT were approved in Germany in 2012-2013, while ICI such as Nivolumab in 2015, however later on in the Results and in the Discussion it seems that data about the effects of TT and ICI have been collected before the 2012, so it is confusing to me. Data were collected also when TT or ICI were used as experimental therapies and not only when they were officially approved as first-line therapies? I think that this could be an important point for the inclusion criteria, if they were used as adjuvant therapies for patients previously treated with chemotherapy or as first-line therapies.
Behind this, Authors state that they investigated the differences in survival of patients treated with new and conventional therapies in Western and Eastern Germany. In general, I think that this focus is not well followed throughout the Results and the Discussion. No evident emphasis was given to this topic, therefore in my opinion Authors should show and comprehensively comment Kaplan-Meier curves comparing the impact on survival of conventional (i.e. chemotherapy and interferon therapy) versus new (TT and ICI) therapies. All the possible comparisons should be analysed and showed by means of graphs representing the survival curves, in order to make it clear the advantage of novel therapies over conventional treatments or the differences between Western and Eastern Germany. This could help the readers to better follow the statistical analysis that the Authors aim to describe.
Moreover, there is a quite relevant disproportion between data analysed from patients between Western and Eastern Germany. Is it statistically significant?
Minor issues:
- in the Abstract, Authors declare to consider only patients with metachronous metastases, while in the results they compare survival data also with synchronous metastasis, please correct the discrepancy.
- In the Introduction, lines 34-38, the Authors introduce in a quite disorganized manner the incidence of melanoma. At the beginning they list North America, Europe, Australia and New Zealand, then they provide the incidence in a different order, not including all the Countries and then introducing Germany. I would suggest to include data for all the Countries they mention or to remove it. And then, focus on Germany alone.
- Line 51: probably the Authors refer to BRAFV600E mutation, sometimes they used V600 or V600E, please standardize it throughout the text.
- Line 42: please provide references for melanoma patients staging.
- Line 47: I would delete immunotherapy since Authors speak here about TT, immunotherapy will be introduced later on.
- In general in the Introduction, Authors provide data about the survival of patients treated with different therapies always in a dissimilar manner, rendering quite difficult to make comparisons. I would suggest to provide the median survival time, expressed in months, or the 1-year survival percentage for each therapy. In any case, however Authors prefer to indicate the data, I would suggest to uniform them, being a work in which the principal aim is to compare survivals.
- Line 86: please add references.
- Line 96: Here the Authors indicate that they aim to assess the differences in overall survival considering also the interferon therapy. However they never mentioned it before, therefore please introduce it (with the relative survival times) when describing conventional therapies such as chemotherapy.
- Line 125: error typing.
- Line 123-132: in my opinion this paragraph is written in a quite confusing manner and it is difficult for the reader to understand the bottom line. Please, make it clearer.
- Figure 2: the figure has a very low resolution. Are the p values indicated for the chemotherapy and interferon therapy correct?
- Paragraph “2.4. Cox regression analysis”: in my opinion, it is very difficult to follow.
Reviewer 2 Report
This paper describes in detail TT and ICI treatment of MM in Germany, 2000-2016, additionally with a focus on regional differences including Western and Eastern Germany.
The paper is well written and the description of the study is adequate but I have several minor issues to be considered before to be published:
- Please check spelling and overall editing.
- Improve the quality of figures and flowcharts. As you have stated it has been implemented and generated in R so it should have a much better quality
- Improve the caption of Fig. 2 to make it more clear.
- Could you please broaden section 5 and suggest what the differences may be between Western and Eastern Germany.
- It is stated that "However, little is known about the implementation of promising new systemic therapies..." line 87. Could describe other works and refine the section related works/state-of-the-art.
Reviewer 3 Report
The authors present their analysis of melanoma treatment and survival in Germany, dividing the patient population into Eastern Germany and Western Germany. They show that targeted therapy produced the best short-term effects on patient survival but immune checkpoint therapy produced better long-term survival in a small set of patients. The authors look at several clinico-pathological characteristics and identify timing of metastasis and melanoma subtype as predictors of patient outcome. The authors demonstrate that these clinico-pathological indicators and availability of treatments differ significantly between the regions, but they go on to suggest that there are independent regional effects on outcome. This is not evident from the data because of the confounding variables described.
The authors should redo their region-specific analyses controlling for the significant predictive variables: metastasis type (metachronous), melanoma subtyping (nodular), and treatment. Without controlling for known predictors, the regional data are uninterpretable. Ideally, the authors would perform a separate statistical analysis for independence to determine whether any hazard can be assigned to region that is not already accounted for by another variable.